# Predictive Attention Transformer: Improving Transformer with Attention Map Prediction

## Abstract

Transformer is a ubiquitous model for natural language processing and has also attracted wide attentions in other domains such as computer vision. The self-attention maps, learned independently for each layer, are indispensable for a transformer model to encode the dependencies among input tokens, however, learning them effectively is still a challenging problem. In this paper, we address this problem and propose a novel approach to improve self-attention through supplementary prediction modules. The underlying assumption is that the attention structures in the current layer should not be completely independent from those in the previous layers and can be better modeled via a convolutional inductive bias. Specifically, we propose Predictive Attention Transformer, which predicts the attention maps through a chain of convolutional neural networks and obtains significant performance gains for various kinds of tasks on top of multiple state-of-the-art models. On GLUE benchmark, the average performances of BERT-Base, BERT-Large, RoBERTa-Large and T5-Base are lifted by 4.1, 2.5, 0.8 and 1.3 points respectively. For ImageNet classification, we achieve significant improvement over multiple backbone models with different capacities.

## 1 Introduction

Transformer (Vaswani et al., 2017) is the state-of-the-art for sequential modeling which achieves superior performances in multiple domains, including natural language understanding (Devlin et al., 2019), image generation (Parmar et al., 2018) and time-series forecasting (Li et al., 2019). The performance of a transformer model largely depends on its capability of inducing reasonable dependencies among input tokens. However, as demonstrated by previous work (Jain & Wallace, 2019), it is difficult for a vanilla attention layer to capture the dependencies effectively without any apriori knowledge. To cope with this problem, recent efforts have tried to address the effectiveness of attention learning, such as concatenating self-attention with CNN layers to obtain a better representation (Bello et al., 2019; Wu et al., 2020), or synthesizing the attention maps directly (Tay et al., 2020). In this paper, we consider another question, *can we improve the learning of attention maps via a dedicated prediction model?* As we will see, it is possible through augmenting the transformer architecture by a chain of convolutional modules for attention map prediction.

For a multi-layer transformer, the self-attention maps in each layer are learned independently, which introduces a huge amount of parameters and hurts the generalization ability. Our motivation is that we can bridge the attention maps from different layers, while a succeeding layer can take the knowledge from previous layers directly to induce a better dependency structure. To this end, we propose Predictive Attention Transformer (PA-Transformer), which guides the learning of attention maps via a chain of convolution-based prediction modules. In each block, PA-Transformer takes all attention maps generated by the previous block as a multi-channel image. Then, by performing 2D-convolution over that image, the attention maps for the current block can be predicted effectively and efficiently. In this way, the general patterns of inter-token dependencies are shared across all blocks, benefiting the generalization ability of a multi-layer Transformer. Meanwhile, the self-attention layer in each block is guided by the predicted attention patterns and can be learned to capture complementary relationships. As shown by a real case of image classification in Figure 1(b), the attention map learned in the second PA-Transformer block correctly highlights the structure of a horse with the help of inherited knowledge from previous layers. Specifically, the convolution-based attention prediction module captures key patterns from a local perspective (probably owing to the

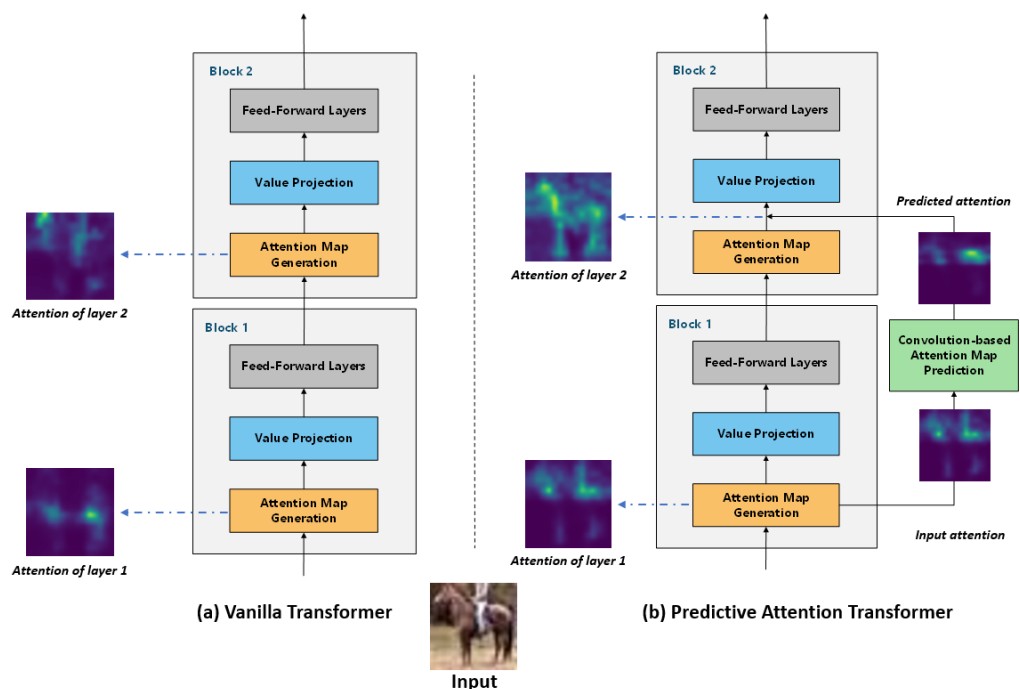

Figure 1: Overview of model architectures with an exemplar case study

convolutional inductive bias), which guides the residual self-attention module to generated better attention maps. In contrast, the vanilla transformer learns each self-attention layer separately and does not produce good attention maps (see Figure 1(a)).

We evaluate the performance of PA-Transformer on plenty of tasks in both natural language and computer vision domains, including text classification, natural language inference, machine translation and image classification. We also apply the generic idea of attention map prediction to other state-of-the-art models such as BERT (Devlin et al., 2019) and AA-ResNet (Bello et al., 2019). The experimental results demonstrate the superiority of PA-enhanced transformers in terms of accuracy, memory and computational costs. In particular, for ImageNet classification task, PA-AA-ResNets achieve strong accuracy improvement on top of AA-ResNet (Bello et al., 2019), a recent SOTA which has already encapsulated self-attention and convolutional layers jointly for image representation. Moreover, we examine the generality of PA-Transformer by incorporating it into BERT-style pre-trained models. Impressively, the average GLUE scores are lifted by 4.1, 2.5, 0.8 and 1.3 points on top of BERT-Base, BERT-Large, RoBERTa-Large and T5-Base respectively, while little extra parameters and computational costs are introduced to the original models.

The contributions of this paper are highlighted as follows.

- In this paper, we propose a novel Transformer architecture augmented by a chain of convolutional attention prediction models. Extensive experiments demonstrate consistent performance enhancement for various natural language and computer vision tasks.

- Ablation study proves the effectiveness of both residual connections among attention maps and convolutional inductive bias to induce better attention patterns. To the best of our knowledge, PA-Transformer is one of the first works that take attention maps as multi-channel input to a deep neural network for explicit modeling. This sheds new lights on the attention mechanism.

- We empirically show that the proposed PA-enhanced method is complementary to existing works on transformer, including those pre-trained by a large corpus and enhanced by CNN layers. Thus, we believe it will have further impacts to more advanced model architectures and a broader range of applications.

## 2 RELATED WORK

Transformer is first introduced by Vaswani et al. (2017) for machine translation and then widely adopted in numerous tasks in natural language (Devlin et al., 2019), computer vision (Parmar et al., 2018; 2019) and time-series (Li et al., 2019) domains. Transformer is solely composed of self-attention and feed-forward layers. It is much more parallelizable than Recurrent Neural Networks (RNNs) and demonstrates extreme superiority in large-scale training scenarios. Most notably, the cutting-edge text representation model, BERT (Devlin et al., 2019), is based on an architecture of deep bidirectional Transformer. After pre-trained on a large-scale language corpus through the "masked language model" (MLM) loss, BERT can be fine-tuned with just one additional output layer to create state-of-the-art performances for a wide range of text-based applications.

The assumption behind Transformer is that the intra-sequence relationships can be captured automatically through self-attention. But in practice, it is questionable if a self-attention layer learns reasonable dependencies among input tokens. There are many endeavors trying to analyze attention maps generated by the attention mechanism. Raganato et al. (2018) analyze the Transformer model for machine translation and show that some attention heads capture certain relations implicitly: lower layers tend to learn more about syntax while higher layers tend to encode more about semantics. As suggested by Tang et al. (2018), the ability of inducing syntactic relations for a Transformer model is weaker than its recurrent neural network counterpart. There is a debate on whether or not the intermediate representations offered by attention mechanisms may be useful to explain the reasons for a model's prediction (Jain & Wallace, 2019; Wiegreffe & Pinter, 2019). Moreover, Synthesizer (Tay et al., 2020) develops a strategy to replace the dot-product attention by synthesized attention maps. It is argued that explicit token-token interaction is not that important. In short, the attention maps learned by existing attention models are far from perfect, which motivates us to propose a dedicated model for extracting attention patterns. Besides, there have been successful attempts to combine convolutional and self-attention layers to enrich image and text representations (Bello et al., 2019; Wu et al., 2020). However, to the best of our knowledge, our work is one of the first that takes attention maps as multi-channel inputs and utilizes a dedicated deep neural network model for explicit modeling. We believe this is a promising direction that deserves more investigations in the future.

Another limitation of Transformer lies in its prohibition for modeling long sequences, as both the memory and computation complexities are quadratic to the sequence length. To address this problem, Kitaev et al. (2020) propose Reformer which utilizes two techniques to improve the efficiency of Transformers: (1) revising dot-product attention with locality-sensitive hashing; (2) replacing residual layers with reversible ones. Moreover, Gehring et al. (2017) leverage an architecture based entirely on convolutional neural networks for sequence to sequence learning, where the number of non-linearities is fixed and independent of the input length. Parmar et al. (2019) apply stand-alone self-attention layers to image classification by restricting the attention operations within a local region of pixels. In addition, there are other directions in the literature for the improvement of a Transformer model, such as relative positional representations (Shaw et al., 2018), adaptive masks for long-range information (Sukhbaatar et al., 2019), tree-based transformer (Shiv & Quirk, 2019), and AutoML-based evolved transformer (So et al., 2019). These works are orthogonal to ours, and most of them can be encapsulated in our framework freely.

## 3 PREDICTIVE ATTENTION TRANSFORMER

### 3.1 OVERVIEW

The representation of a text sequence can be written as $\mathbf{X} \in \mathbb{R}^{N \times C}$, where $N$ denotes the sequence length and $C$ is the dimension size. For an image representation, the conventional shape is $(H, W, C)$, where $H, W$ and $C$ denote height, width and channel size of the image respectively. In order to apply a standard Transformer to the image representation, we flatten its shape as $\mathbf{X} \in \mathbb{R}^{N \times C}$, where $N = HW$ and each pixel serves as an individual token in the Transformer.

A standard Transformer block is composed of a self-attention layer and two position-wise feed-forward layers, while each attention map is generated by a self-attention layer separately without sharing information with each other. We argue that a single and independent self-attention layer is not generalizable to capture the underlying dependencies among tokens. To alleviate this prob-

lem, we propose a convolution-based prediction module that calculates attention maps for the current layer based on the attention map from the previous layer. Our conjecture is that this module would predict effective attention maps in the guidance of generalized attention patterns from previous layers. Thus, the self-attention module in the current layer could dedicate itself to incorporate layer-specific knowledge into residual attention maps.

We name the transformer network with proposed augmentation as Predictive Attention Transformer (PA-Transformer), the architecture of which is illustrated in Figure 1(b). Each PA-Transformer block consists of three modules as a standard transformer, namely *Attention Map Generation*, *Value Projection* and *Feed-Forward Layers*. In addition, there are *Convolution-based Attention Prediction* modules, which take the attention maps from the previous block as input to predict the ones in the next block through convolutional neural networks. Note that *Attention Map Generation* and *Value Projection* are two sub-modules of a standard self-attention layer. We omit layer norm and residual connections in the figure for brevity. In the rest of this section, we will introduce the details of each module separately.

## 3.2 ATTENTION MAP GENERATION

Given the input representation $\mathbf{X}$, the attention map can be calculated as follows. First, we compute the query and key matrices for each attention head through linear projections, *i.e.*, $\mathbf{Q} = \mathbf{X}\mathbf{W}^Q$, $\mathbf{K} = \mathbf{X}\mathbf{W}^K$, where $\mathbf{Q}$ and $\mathbf{K}$ denote query and key matrices respectively, $\mathbf{W}^Q$ and $\mathbf{W}^K$ are linear projection parameters. Then, the attention map is derived by a scaled dot-product operation:

$$\mathbf{A} = \text{Attention}(\mathbf{X}) = softmax(\frac{\mathbf{Q}\mathbf{K}^\top}{\sqrt{d}}) \tag{1}$$

Here $\mathbf{A}$ denotes the attention map and $d$ is the hidden dimension size. To inject sequential information into the model, we incorporate positional encoding to the input representation. The positional encoding can be either absolute or relative, and we follow the original implementation for different backbone models. For absolute positional encoding (Vaswani et al., 2017), it is added to the input representation $\mathbf{X}$ directly. For relative positional representation (Shaw et al., 2018), the attention formulation can be re-written as:

$$\mathbf{A} = \text{Attention}(\mathbf{X}) = softmax(\frac{\mathbf{Q}\mathbf{K}^\top}{\sqrt{d}} + \mathbf{R}) \tag{2}$$

where $\mathbf{R} = \{\mathbf{r}_{ij}\}$ is the matrix of relative positional encoding. For text data, we have $\mathbf{r}_{ij} = \mathbf{q}_i^T \mathbf{e}_{i-j}$, where $\mathbf{e}_{i-j}$ is a trainable embedding vector in terms of the relative index of two tokens. For image data, we adopt two separate embedding vectors for height and width (Bello et al., 2019).

$$\mathbf{r}_{ij} = \mathbf{q}_i^\top \mathbf{e}^H_{h(j)-h(i)} + \mathbf{q}_i^\top \mathbf{e}^W_{w(j)-w(i)} \tag{3}$$

where $\mathbf{q}_i$ is the query representation for the $i^{th}$ pixel, $\mathbf{e}^H$ and $\mathbf{e}^W$ represent for trainable embedding vectors of height and width respectively, $h(i)$ and $h(j)$ are the height indices for the $i^{th}$ and $j^{th}$ pixels, and $w(\cdot)$ denotes the index in width.

## 3.3 CONVOLUTION-BASED ATTENTION PREDICTION

In a vanilla Transformer, the attention maps are calculated by a single dot-product operation. However, it may not be effective for constructing dependencies among tokens, since each attention map is generated independently without sharing information with each other. To tackle this limitation, we propose a convolution-based prediction module, which calculates attention maps for a new block by applying a convolutional neural network to existing ones. The prediction module is expected to induce effective attention patterns in the guidance of generic knowledge. Thus, the self-attention module can dedicate itself to capture residual knowledge for a specific layer.

Assume there are $K$ heads in each layer, then we have $K$ attention map outputs from a *Attention Map Generation* module. They construct a tensor $A \in \mathbb{R}^{N \times N \times K}$ ($N$ is the sequence length), which can be viewed as a $N \times N$ image with $K$ input channels. On the basis of this input, we adopt several 2D-convolutional layers with $3 \times 3$ kernels to predict the attention maps for the next block. The output channel is also set to be $K$, so the attention maps of all heads can be generated jointly.

We apply a ReLU activation after each 2D-convolutional layer to provide non-linearity and sparsity. Finally, the predicted attention map is combined with that learned by dot-product attention:

$$\mathbf{A} = softmax(\alpha \cdot \text{CNN}(\mathbf{A}_{pre\_logits}) + (1 - \alpha) \cdot \text{Attention}(\mathbf{X})_{logits}) \tag{4}$$

where $\mathbf{A}_{pre\_logits}$ is the matrix of attention logits (before softmax) from the previous layer; $\text{Attention}(\mathbf{X})_{logits}$ is the attention matrix of logits calculated by the current self-attention layer; $\alpha \in [0, 1]$ is a hyper-parameter to balance the importance of two branches. In our experiments, $\alpha$ is chosen empirically on the validation set. We will analyze the impact of different $\alpha$ values in Section 5. We do not apply convolution-based attention prediction to the first transformer block.

### 3.4 Value Projection and Feed-Forward layers

Given the attention map $\mathbf{A}$, the rest of a PA-Transformer block includes value projection and position-wise feed-forward layers that are identical to a standard transformer block. The value projection layer can be formulated as:

$$\mathbf{H}_i = \mathbf{A}_i \mathbf{X} \mathbf{W}_i^V, \quad \mathbf{H} = (\mathbf{H}_1 \oplus \mathbf{H}_2 \oplus ... \oplus \mathbf{H}_K) \mathbf{W}^O \tag{5}$$

where $\mathbf{A}_i$ is the attention map for the $i^{th}$ head, $\mathbf{W}_i^V$ is the parameter of value projection, and $\mathbf{H}_i$ is the corresponding representation generated by value projection. Afterwards, the representation of each head is concatenated (denoted by $\oplus$) and fed into a linear projection layer with parameter $\mathbf{W}^O$. At last, the block is followed by two position-wise feed-forward layers:

$$\text{PA-Transformer}(\mathbf{X}) = \text{ReLU}(\mathbf{H}\mathbf{W}_1 + \mathbf{b}_1)\mathbf{W}_2 + \mathbf{b}_2 \tag{6}$$

Conventionally, the dimension of $\mathbf{W}_1$ is four times of $\mathbf{W}^O$ and $\mathbf{W}_2$ to form a bottleneck structure.

## 4 Experiments

### 4.1 Natural Language Processing

First, we start from scratch training scenarios. Then, we conduct experiments of fine-tuning based on pre-trained models. This is a practical setting, as it will be exhaustive to test all model variants by pre-training them from scratch.

#### 4.1.1 Scratch Training Scenarios

We consider three NLP tasks which are commonly adopted for the evaluation of scratch training: sentiment classification, natural language inference and machine translation.

**Datasets** We adopt SST (Socher et al., 2013) and Yelp (Zhang et al., 2015) datasets for sentiment classification, SNLI (Bowman et al., 2015) for natural language inference, and IWSLT'14 German-English (De-En) (Cettolo et al., 2014) for machine translation. Max token length is set to be 64 for both SST and SNLI, 256 for Yelp, and 4096 for IWSLT'14 De-En. Detailed descriptions are available in Appendix A.

**Models.** Existing works (Bello et al., 2019) have showed the effectiveness of convolutional neural network to be used in parallel with self-attention in a multi-branch structure. Our work leverages CNN for reasoning on the attention maps, which is orthogonal to existing ones. To examine the complementarity, we take a multi-branch network architecture that concatenates self-attention and convolutional layers jointly as another baseline model. This architecture is named as Conv-Transformer, and its predictive counterpart is called PA-Conv-Transformer (see Figure 4 in the appendix). To examine the effectiveness of the proposed method, we compare the following model variants: (1) *CNN* contains $M$ layers of 1D-convolution of kernel size 3. (2) *Transformer* is stacked by $M$ standard Transformer blocks (Figure 1(a)). (3) *PA-Transformer* is stacked by $M$ Predictive Attention Transformer blocks as visualized in Figure 1(b). (4) *Conv-Transformer* leverages a multi-branch architecture that concatenates self-attention and convolutional layers in each Transformer block. (5) *PA-Conv-Transformer* is stacked by $M$ Predictive Attention Convolutional Transformer blocks (Figure 4). We set $M = 3$ by default except for the machine translation task, where we follow the implementation of Joulin et al. (2017) and set the block number as 6 for both the encoder and

| Model | #Params | #FLOPs (SST) | SST | Yelp | SNLI | De-En |
|---|---|---|---|---|---|---|
| CNN | 2.77M | 317.38M | 43.56 ± 0.43 | 62.09 ± 0.72 | – | – |
| Transformer | 2.79M | 338.86M | 47.64 ± 0.70 | 63.23 ± 0.39 | 83.22 ± 0.39 | 31.64 |
| **PA-Transformer** | 2.79M | 349.10M | **48.60 ± 0.41** | **63.90 ± 0.18** | **84.63 ± 0.29** | **32.51** |
| Conv-Transformer | 2.58M | 321.34M | 49.74 ± 0.55 | 64.21 ± 0.09 | 85.40 ± 0.61 | 32.24 |
| **PA-Conv-Transformer** | 2.58M | 331.39M | **50.66 ± 0.64** | **64.75 ± 0.30** | **86.28 ± 0.25** | **32.48** |

Table 1: Performance for natural language processing tasks

| Model | Avg | CoLA | SST-2 | MRPC | STS-B | QQP | MNLI-m/-mm | QNLI | RTE | WNLI |
|---|---|---|---|---|---|---|---|---|---|---|
| BERT-Base | 77.4 | 51.7 | 93.5 | 87.2/82.1 | 86.7/85.4 | 91.1/89.0 | 84.3/83.7 | 90.4 | 67.2 | 65.1 |
| **PA-BERT-Base** | **81.5** | **59.8** | **93.7** | **88.9/90.8** | **89.3/89.2** | **91.4/88.3** | **84.8/85.2** | **92.0** | **68.6** | 65.1 |
| BERT-Large | 80.5 | 60.5 | 94.9 | 89.3/85.4 | 87.6/86.5 | 92.1/89.3 | 86.8/85.9 | 92.7 | 70.1 | 65.1 |
| **PA-BERT-Large** | **83.0** | **63.1** | **95.4** | **90.4/88.9** | **88.9/88.0** | **92.4/89.9** | **87.7/86.2** | **93.5** | **70.9** | 65.1 |
| RoBERTa-Large | 83.1 | 63.8 | 96.3 | 91.0/89.4 | 72.9/90.2 | 92.7/90.1 | 89.5/89.7 | 94.2 | 84.2 | 65.1 |
| **PA-RoBERTa-Large** | **83.9** | **65.4** | **96.5** | **91.8/90.6** | **73.6/90.3** | **93.0/90.1** | **90.3/89.7** | **95.0** | **85.2** | 65.1 |
| T5-Base | 83.2 | 51.1 | **95.2** | 90.7/87.5 | 89.4/88.6 | 72.6/89.4 | **87.1/86.2** | **93.7** | 80.1 | – |
| **PA-T5-Base** | **84.5** | **53.3** | 92.8 | **92.4/89.5** | **89.6/89.1** | **89.4/91.5** | 86.0/85.1 | 92.9 | **80.7** | – |
| T5-Base (Dev) | 83.5 | 53.1 | 92.2 | 92.0/88.7 | 89.1/88.9 | 88.2/91.2 | 84.7/**85.0** | 91.7 | 76.9 | – |
| Synthesizer-T5-Base (Dev) | 84.1 | 53.3 | 92.2 | 91.2/87.7 | 89.3/88.9 | 88.6/91.4 | 85.0/84.6 | 92.3 | **81.2** | – |
| **PA-T5-Base (Dev)** | **84.4** | **53.6** | **92.4** | **92.2/89.1** | **89.7/89.1** | **89.7/91.7** | **85.4**/84.9 | **92.5** | 80.7 | – |

Table 2: Comparison of different model backbones on GLUE benchmark.

decoder networks. We only apply PA-Transformer architecture to the encoder while its application to decoder is considered as future work. We use one convolutional layer in each prediction module. More analysis can be found in Section 5.

**Settings.** For sentiment classification and natural language inference tasks, the hidden dimension size is set to be 256 with 8 heads, and the first feed-forward layer in each block has a hidden size of 1024. An initial $1 \times 1$ convolutional layer is applied to the input embedding. In Conv-Transformer , the hidden size is one half of the total dimension for both convolutional and self-attention branches, and the self-attention branch has 4 heads. We apply dense connections between different layers similar to DenseNet (Huang et al., 2017). The word embedding vectors are initialized by GloVe (Pennington et al., 2014) and fixed during training. For machine translation, we follow a recent work (Wu et al., 2020) that removes the bottleneck structure and set the hidden size as 160 for all layers. Besides, we utilize 10K joint byte pair encoding (BPE) (Sennrich et al., 2016) vocabulary in lower case. The hyper-parameters are chosen empirically on the validation set of each dataset. Specifically, the value of $\alpha$ is selected by grid search from a search space of $\{0.01, 0.025, 0.05, 0.075, 0.1, 0.15, 0.2, 0.3, ..., 0.9\}$ on the validation set. Finally, we set it as 0.1, 0.1, 0.5 and 0.1 for both PA-Transformer and PA-Conv-Transformer on SST, Yelp, SNLI and IWSLT'14 De-En datasets respectively. All models are trained by the Adam optimizer (Kingma & Ba, 2014). More settings are described in Appendix B.2.1.

**Results.** The performances of different models on four natural language processing datasets are summarized in Table 1. For the first three datasets, the numbers are obtained by running each model five times with random seeds, and their average accuracy and standard deviation are reported. For De-En machine translation, we leverage a default seed and report the BLUE score. Note that the number of parameters and FLOPs changes for each dataset, and we just report the ones on SST for inference. As shown in Table 1, PA-Transformer outperforms vanilla Transformer significantly on all datasets. Consistent with previous findings (Wu et al., 2020), we observe steady improvements by concatenating convolutional and self-attention layers for text representation (refer to Conv-Transformer). Furthermore, as indicated by PA-Conv-Transformer, incorporating convolution-based attention prediction modules into Conv-Transformer brings supplementary benefits. The comparisons are fair since all models have similar numbers of parameters and FLOPs.

### 4.1.2 FINE-TUNING FROM PRE-TRAINED MODELS

Pre-trained language models like BERT (Devlin et al., 2019) become popular in recent years. These models are based on the bi-directional transformer architecture and pre-trained using Masked Lan-

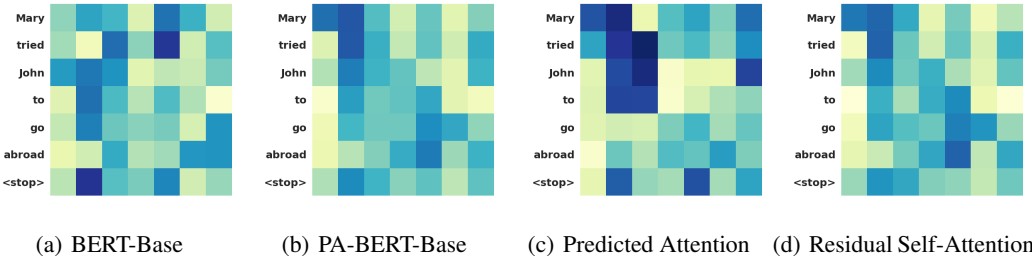

| (a) BERT-Base | (b) PA-BERT-Base | (c) Predicted Attention | (d) Residual Self-Attention |

Figure 2: Visualization of attention maps for an exemplar case study

guage Model (MLM) loss on a large corpus. An interesting question is that if the proposed attention prediction module is still useful when fine-tuned from a pre-trained model checkpoint. To answer this question, we choose GLUE benchmark (Wang et al., 2018) for an empirical study.

**Settings.** The encoder network of BERT consists of multiple transformer blocks. In PA-BERT, We replace each transformer block with PA-Transformer proposed in this paper. We load the pre-trained checkpoints of BERT-Base, BERT-Large, RoBERTa-Large and T5-Base directly and fine-tune them for different downstream tasks. For PA-BERT, the additional parameters are initialized randomly and trained jointly during fine-tuning. The hyper-parameters are chosen on validation sets and reported in Appendix B.2.2. Finally, we make predictions on the test data and send the results to GLUE online evaluation service.

**Results.** The comparison between BERT-style models are shown in Table 2. PA-BERTs perform generally better than vanilla BERTs on mutiple downstream tasks. Specifically, PA-BERT-Base, PA-BERT-Large and PA-RoBERTa-Large achieve average scores of 81.5, 83.0 and 83.9 on the GLUE test set, increasing 4.1, 2.5 and 0.8 points based on BERT-Base, BERT-Large and RoBERTa-Large respectively. We also compare vanilla Transformer, Synthesizer and PA-Transformer architectures on T5-Base. Note that the corresponding average scores do not count for the WNLI dataset following the baseline setting (Tay et al., 2020). Impressively, PA-T5-Base outperforms Synthesizer after being fine-tuned on each task individually from T5-Base checkpoint, whereas Synthesizer is pre-trained from scratch and fine-tuned in a multi-task framework. This is advantageous since PA-Transformer can be applied to improve any pre-trained transformer models without exhaustive pre-training again. In addition, it is worth mentioning that PA-BERT-Base boosts the score by 8.1 on CoLA dataset, indicating its superior generalization ability for small datasets.

**Analysis.** We compare the attention maps generated by BERT-Base and PA-BERT-Base for more insights. We find that the PA-BERT-Base model can better highlight the key tokens and their relationships for making the final decision. For instance, we have a case study in Figure 2 by visualizing the attention maps from the last layer. The sentence is "Mary tried John to go abroad.", and the task is to check the grammatical correctness. As shown in Figure 2(a), BERT-Base only focuses on verbs and stop signs, leading to a mis-classification result. In contrast, PA-BERT-Base (Figure 2(b)) learns to attend to the relationships between "tried" and "John", which correctly captures the error part and gives the right answer. In addition, we visualize the attention maps generated by the convolution-based prediction layer and the residual self-attention layer in Figure 2(c) and (d) respectively. The design of generalizing inter-token patterns through convolutional inductive bias seems to be beneficial for emphasizing problematic parts, whereas self-attention captures complementary relationships, for example, between the words "go" and "abroad".

## 4.2 IMAGE CLASSIFICATION

### 4.2.1 CIFAR

**Settings.** The model architecture used for CIFAR image classification is similar to that introduced in Section 4.1 except for $3 \times 3$ 2D-convolution kernels are applied to image representation. All models are stacked by 12 blocks with 64 filters at the beginning. For every 3 blocks, we reduce the image size via a $2 \times 2$ average pooling layer and double the hidden size at the same time. Finally, we apply

| Model | #Params (CIFAR-10) | #FLOPs | CIFAR-10 | CIFAR-100 |
|---|---|---|---|---|
| CNN | 5.36M | 1.11G | 95.90 | 78.17 |
| Conv-Transformer | 4.92M | 1.05G | 95.82 | 79.86 |
| **PA-Conv–Transformer** | 4.92M | 1.14G | **96.22** | **80.21** |

Table 3: Accuracy on CIFAR image classification datasets

| Model | #Params | #GFLOPs | Top-1 | Top-5 |
|---|---|---|---|---|
| ResNet-34 | 21.8M | 7.4 | 73.79 | 91.43 |
| AA-ResNet-34 | 20.7M | 7.1 | 74.33 | 91.92 |
| **PA-AA-ResNet-34** | 20.7M | 7.9 | **74.90** | **92.20** |
| ResNet-50 | 25.6M | 8.2 | 75.99 | 93.00 |
| AA-ResNet-50 | 25.8M | 8.3 | 77.15 | 93.52 |
| **PA-AA-ResNet-50** | 25.8M | 8.7 | **77.55** | **93.81** |
| ResNet-101 | 44.5M | 15.6 | 77.40 | 93.65 |
| AA-ResNet-101 | 45.4M | 16.1 | 78.31 | 94.16 |
| **PA-AA-ResNet-101** | 45.4M | 17.2 | **78.49** | **94.23** |

Table 4: Accuracy on ImageNet dataset

a global average pooling layer and feed the flattened representation to the classifier. Note that the memory is exhaustive for the original image size ($32 \times 32$). Thus, we only leverage self-attention after the image size is reduced to $8 \times 8$. We set $\alpha = 0.01$ empirically.

**Results.** As shown in Table 3, Conv-Transformer is on par with CNN on CIFAR-10 and performs significantly better on the more difficult dataset, CIFAR-100. Conv-Transformer is a strong baseline, as it matches the performance of 7M DenseNet (Huang et al., 2017) on both CIFAR-10 and CIFAR-100 datasets with less than 5M parameters. After adding the attention prediction module, PA-Conv-Transformer boosts the performances on two datasets consistently with comparable parameters and computations, demonstrating the superiority of our approach.

### 4.2.2 IMAGENET

AA-ResNet (Bello et al., 2019) proved that traditional CNN models could benefit from attention mechanisms. One may curious about if our revised attention model would bring additional advantages. Hence, we take AA-ResNet as the backbone model for evaluation on ImageNet.

**Settings.** We follow the experimental protocol of AA-ResNet with standard ResNet architectures (He et al., 2016). We set $\alpha = 0.5$ for PA-AA-ResNet and adopt one convolutional layer for each attention prediction module. All models are trained by 100 epochs on 8 TESLA V100 GPUs. Major hyper-parameters are as follows: optimizer is SGD with momentum 0.9, batch size is 32 per worker, weight decay is 1e-4. For the first 5 epochs, the learning rate is scaled linearly from 0 to 0.128, and then it is divided by 10 at epoch 30, 60, 80 and 90 respectively.

**Results.** As shown in Table 4, AA-ResNets outperform corresponding ResNet models by a large margin. The proposed PA-AA-ResNets further boost the top-1 accuracy by 0.73%, 0.51% and 0.23% on top of AA-ResNet-34, -50 and -101 respectively. These numbers are statistically significant under 95% confidence level. Arguably, the performance lifts are owing to better attention maps calculated by the proposed framework of convolution-based predictive attention.

## 5 ANALYSIS

**Effectiveness of convolutional layers.** Our default configuration utilizes one convolutional layer for an attention prediction module. To examine the effectiveness of convolutional neural networks and the impact of layer number, we take layer number as a hyper-parameter and compare the results of different settings. When layer number is set to zero, the setting equals to a vanilla Transformer with residual connections (referred to as "Transformer with RC"). The performances of vanilla

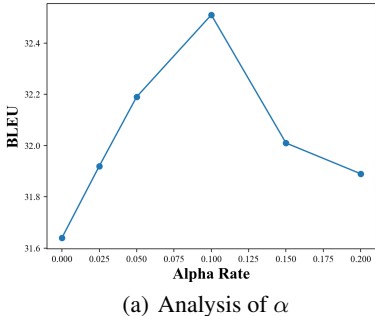

(a) Analysis of $\alpha$

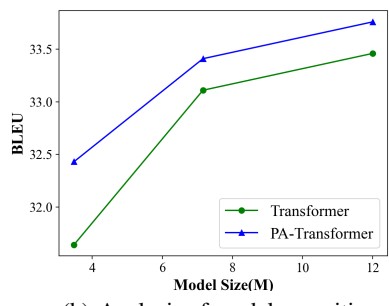

(b) Analysis of model capacities

Figure 3: Analysis on IWSLT'14 De-En dataset

| Model | SST | SNLI | De-En | Model | ImageNet |
|---|---|---|---|---|---|
| Transformer | 47.64 | 83.22 | 31.64 | AA-ResNet-34 | 74.33 |
| Transformer with RC | 48.14 | 83.81 | 32.30 | AA-ResNet-34 with RC | 74.36 |
| 1-Layer PA-Transformer | 48.62 | 84.63 | **32.51** | 1-Layer PA-AA-ResNet-34 | **74.90** |
| 2-Layer PA-Transformer | **48.82** | **84.78** | 31.67 | 2-Layer PA-AA-ResNet-34 | 74.35 |
| 3-Layer PA-Transformer | 48.22 | 84.65 | 31.55 | 3-Layer PA-AA-ResNet-34 | – |

Table 5: Analysis of PA-Transformer  |  Table 6: Analysis of AA-ResNet-34

Transformer, Transformer with RC, and PA-Transformers consisting of different convolutional layers are reported in Table 5. We can see that the residual connection is beneficial, and we generally achieve strong performances on all datasets through one convolutional layer. Similar trends are observed in the ImageNet experiments, where we analyze the effectiveness of residual connections and convolution-based prediction modules in Attention Augmented ResNet-34 architecture. As shown in Figure 6, the best performance is obtained with one predictive convolutional layer.

**Sensitivity of $\alpha$.** In Equation (4), there is a ratio $\alpha \in [0, 1]$ to combine the attention maps generated by self-attention and convolution-based prediction. Figure 3(a) and Figure 5 (in the appendix) visualize the test accuracy of different values of $\alpha$ on IWSLT'14 De-En and SNLI datasets. Note that $\alpha = 0$ corresponds to a vanilla Transformer. Although the best $\alpha$ varies in different datasets, the performance gains are stable for multiple choices of $\alpha$. Thus, one can easily find a reasonable value by searching on the validation set. In addition, we want to point out that the value of $\alpha$ is not necessarily corresponding to a relative importance, but also has an effect of normalization. In the future work, we would like to explore advanced strategies to get rid of this hyper-parameter.

**Impact of model capacity.** We further compare Transformer and PA-Transformer with different parameter sizes on De-En dataset in Figure 3(b). We follow the setting in Wu et al. (2020) to take three models with roughly 3M, 6M and 9M parameters. As illustrated by the figure, PA-Transformer scales well with the growth of parameter size and outperforms vanilla Transformer consistently with different model capacities. Experiments on ImageNet (Table 4) demonstrate a similar trend.

## 6 CONCLUSION

In this paper, we propose a novel transformer model, PA-Transformer, which facilitates the learning of attention maps by a chain of convolutional neural networks. It demonstrates superior performances on various tasks in NLP and CV domains. Future works are considered in three aspects. First, we aim to incorporate this idea into the decoder network and evaluate its effectiveness. Second, we would like to explore its applications for more tasks and domains, such as question answering, object detection and time-series forecasting. Last but not least, we plan to develop a more automatic strategy for choosing or replacing the hyper-parameter $\alpha$.

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

## A DATASETS

**SST.** The Stanford Sentiment Treebank (SST) dataset (Socher et al., 2013) contains more than 10,000 movie reviews collected from *rottentomatoes.com* website, while each piece of review is labeled by one of five fine-grained categories of sentiment polarity. We utilize the standard train/dev/test split of this dataset.

**Yelp.** The Yelp dataset (Zhang et al., 2015) contains business reviews from *yelp.com* which are also classified into five polarities. We select 10% of the training data randomly for validation.

**SNLI.** The Stanford Natural Language Inference (SNLI) dataset[1] (Bowman et al., 2015) contains 570k human-written English sentence pairs manually labeled for balanced classification with the following labels: entailment, contradiction, and neutral. We follow the original split of the released data.

**IWSLT'14 De-En** The IWSLT'14 German-English (De-En) machine translation dataset (Cettolo et al., 2014) comes from translated TED talks. The dataset contains roughly 153K training parallel sentences, 7K development sentences, and 7K test sentences. We apply BPE Sennrich et al. (2016) to the source language corpus and the target language corpus separately and the final vocabulary sizes of German and English are 8848 and 6632, respectively.

**GLUE** The GLUE (General Language Understanding Evaluation) benchmark[2] contains several different NLP tasks in different fields. In STS-B, each pair of sentences is annotated by a similarity score from 1 to 5, and the task is to predict the target score by regression. All other tasks are supervised classification ones, where MRPC and QQP are similarity and paraphrase tasks; CoLA and SST-2 are single-sentence classification tasks; MNLI, RTE, and QNLI are pairwise inference tasks. Such a wide range of NLP tasks can be used to assess the generalization ability of different methods.

**CIFAR.** CIFAR[3] contains colored natural images with $32 \times 32$ pixels. CIFAR-10 consists of images drawn from 10 classes and CIFAR-100 contains 100 classes in total. The training and test sets contain 50,000 and 10,000 images respectively, and 5,000 training images are hold out for validation.

**ImageNet.** The ImageNet 2012 classification dataset[4] comes from ImageNet Large Scale Visual Recognition Challenge (Russakovsky et al., 2015), which consists of 1000 classes with 1.28 million training images and 50k validation images. One can submit the inference results to online evaluation service to get the test accuracy.

---

[1] https://nlp.stanford.edu/projects/snli/
[2] https://gluebenchmark.com/
[3] https://www.cs.toronto.edu/~kriz/cifar.html
[4] http://www.image-net.org/

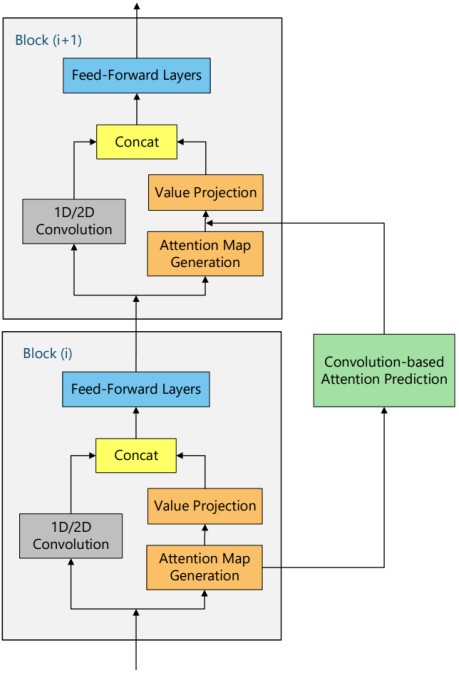

Figure 4: Architecture of PA-Conv-Transformer

## B DETAILED SETTINGS FOR REPRODUCTION

### B.1 PA-CONV-TRANSFORMER

The architecture of Predictive Attention Convolutional Transformer (PA-Conv-Transformer) is illustrated in Figure 4. Compared to a non-convolutional counterpart, we use one half of dimension for the convolutional layer and the other half for the self-attention layer. The head number of self-attention is set as 4.

### B.2 NATURAL LANGUAGE PROCESSING

#### B.2.1 SCRATCH TRAINING

The Natural Language Inference (NLI) task takes a pair of sentences as input. Similar to BERT-style implementation, we concatenate two sentences into a single one by putting a special token in the middle and adding a field embedding vector to the input representation. Based on this design, we can apply the same architecture as sentiment classification to this task.

We train SST and Yelp sentiment classification datasets for 10 epochs with cosine learning rate decay (Loshchilov & Hutter, 2017), where the initial learning rate is set as 4e-4 and the end learning rate is set to be 1e-6. For SNLI dataset, the initial learning rate is set as 4e-4 and is divided by 10 every 10 epochs until 30 epochs are finished. For all datasets, the batch size is 64 and $l_2$ norm is 2e-6. We adopt 0.4 and 0.2 dropout ratios for the embedding and fully-connected layers respectively. These hyper-parameters are chosen empirically on the validation set using a vanilla Transformer. To avoid exhaustive search, we do not tune the hyper-parameters on different datasets and model variants, except for the linear combination ratio $\alpha$ in PA-Transformer and PA-Conv-Transformer. We adopt relative positional encoding in Shaw et al. (2018).

We train IWSLT'14 De-En dataset using Adam optimizer (Kingma & Ba, 2014) with $\beta_1 = 0.9$, $\beta_2 = 0.98$ and an inverse square root learning rate scheduling with linear warmup. The learning rate is 5e-4 and the warmup step is set to 4000 as in Vaswani et al. (2017). Also, we use dropout rate $p = 0.2$ and label smoothing $\epsilon = 0.1$. we apply early stopping to the training procedure

| Task | Training Epochs | Batch Size | Learning Rate | Adam Epsilon | Dropout Rate | $\alpha$ |
|------|-----------------|------------|----------------|---------------|---------------|----------|
| CoLA | 3 | 8 | 2e-5 | 1e-8 | 0.1 | 0.2 |
| SST-2 | 5 | 8 | 1e-5 | 1e-8 | 0.1 | 0.1 |
| MRPC | 2 | 8 | 2e-5 | 1e-8 | 0.1 | 0.2 |
| STS-B | 3 | 8 | 2e-5 | 1e-8 | 0.1 | 0.4 |
| QQP | 3 | 16 | 2e-5 | 1e-8 | 0.1 | 0.2 |
| MNLI | 3 | 16 | 2e-5 | 1e-8 | 0.1 | 0.4 |
| QNLI | 3 | 16 | 2e-5 | 1e-8 | 0.1 | 0.4 |
| RTE | 2 | 8 | 2e-5 | 1e-8 | 0.1 | 0.2 |
| WNLI | 2 | 8 | 2e-5 | 1e-8 | 0.1 | 0.2 |

Table 7: Detailed hyper-parameter settings for GLUE fine-tuning.

with a patience of 5 epochs. In the evaluation phase, we average the final 10 checkpoints and conduct beam search with size 5. We adopt absolute positional encoding according to the original implementation (Vaswani et al., 2017).

### B.2.2 PRE-TRAINING

As introduced in Devlin et al. (2019), BERT-Base and PA-BERT-Base have 12 layers and 12 attention heads with hidden dimension 768. BERT-large and Conv-BERT-large have 24 layers and 16 heads, while hidden dimension for each intermediate layer is set as 1024. The hidden dimension of the final fully-connected layer before Softmax is set to be 2000. We download the officially released checkpoints of BERT-Base[5] and BERT-Large[6], and initialize the additional parameters for PA-BERT-Base and PA-BERT-Large randomly.

Meanwhile, we also conducted a set of comparative tests with RoBERTa-Large (Liu et al., 2019) and T5-Base (Raffel et al., 2019) as the backbone models. We add the idea of predictive attention map to the attention structure of these models, named as PA-RoBERTa-Large and PA-T5-Base respectively. For RoBERTa and PA-RoBERTa, they have 24 layers with 16 attention heads. The total hidden size of all heads is 1024, and the hidden dimension of the final fully-connected layer is 4096. Following (Raffel et al., 2019), we use NLP library Transformers(Wolf et al., 2019) implemented by the huggingface team to implement the base version of T5, which has 220 million parameter. In order to fine-tune the tasks in the glue benchmark, we downloaded the official pre-trained parameters for RoBERTa-Large[7] and T5-Base[8] as our start checkpoints.

We use the Adam optimizer (Kingma & Ba, 2014) with epsilon 1e-8. The dropout rate is set as 0.1 empirically. We used grid search to optimize the values of hyper-parameters on validation data. We search the learning rate in {1e-4, 1e-5, 2e-5}, batch size in {8, 16}, training epochs in {2, 3, 5} and $\alpha$ of equation 4 in {0.1, 0.2, 0.4}. We find that the following setting is the best choice for most tasks: learning rate 2e-5, batch size 8, training epoch number 3 and $\alpha = 0.2$. The specific hyper-parameter for each task is shown in Table 7.

### B.3 IMAGE CLASSIFICATION

### B.3.1 CIFAR

We have two bottleneck feed-forward layers on top of each CNN/Conv-Transformer/PA-Conv-Transformer block. There are dense connections between different layers like DenseNet (Huang et al., 2017). The training batch size is set as 50 for all models. We adopt cosine learning rate schedule according to Pham et al. (2018).

$$l = l_{min} + 0.5 \cdot (l_{max} - l_{min})(1 + cos(\pi T_{cur}/T)) \tag{7}$$

---

[5]https://storage.googleapis.com/bert_models/2018_10_18/uncased_L-12_H-768_A-12.zip
[6]https://storage.googleapis.com/bert_models/2018_10_18/uncased_L-24_H-1024_A-16.zip
[7]https://dl.fbaipublicfiles.com/fairseq/models/roberta.large.tar.gz
[8]https://console.cloud.google.com/storage/browser/t5-data/pretrained_models/base/

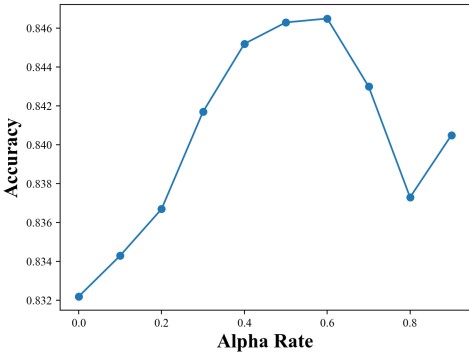
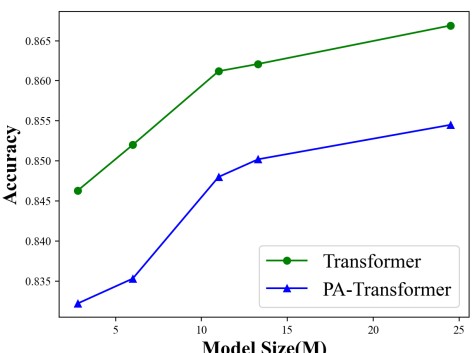

Figure 5: Different values of $\alpha$ on SNLI     Figure 6: Different model capacities on SNLI

Where $l$ is the learning rate and $T_{cur}$ is the epoch ID in the current cycle. We set $l_{max} = 0.05$ and $l_{min} = 0.0001$. For CIFAR-10, the value of $T$ is set as 10 in the first cycle and is multiplied by a factor of 2 at the end of each cycle. The dropout rates of convolutional, self-attention and MLP layers are set to be 0.2, 0.2 and 0.2 respectively; $l_2$ norm is set as 1e-4. Each model is trained by 630 epochs which consist of six complete cycles. For CIFAR-100, the value of $T$ is 20 in the first cycle and is multiplied by a factor of 2 at the end of each cycle. The dropout rates of convolutional, self-attention and MLP layers are set as 0.2, 0.0 and 0.2 respectively; $l_2$ norm is set as 1e-4. Each model is trained by 620 epochs which consist of five complete cycles. Cutout size (DeVries & Taylor, 2017) is set as 8 for both CIFAR-10 and CIFAR-100 datasets.

### B.3.2    IMAGENET

We follow a common strategy (Szegedy et al., 2016) for data augmentation. For ResNet (He et al., 2016), we adopt the implementation in tensorflow CNN benchmark[9]. For AA-ResNet, we modify the ResNet by augmenting 3x3 convolution with self-attention. specifically, we apply attention augmentation to each residual block in the last 3 stages – when the shapes of activation maps become 28x28, 14x14 and 7x7. We refer to Bello et al. (2019) for more details. The implementation is from the official repository[10], and we simply add attention map prediction module to the code base. If not specified, we adopt the same setting with AA-ResNet, e.g. $k = 2$ and $v = 0.2$. We conduct hyper-parameter search for the value of $\alpha$ and the number of convolutional layers. The final performance of PA-AA-ResNet is obtained when the number of convolution layer is 2, and $\alpha$ is set to be 0.5.

## C    ANALYSIS

### C.1    ANALYSIS ON SNLI

The analysis of different $\alpha$ values and model capacities on SNLI dataset is shown in Figure 5 and 6 respectively. All the result numbers are average of 5 different runs with random seeds. We search for multiple layer numbers in $\{3, 6, 9, 12\}$ and hidden dimension sizes in $\{256, 384, 512, 768\}$, and find that the following settings are the best among competitors of similar model sizes: (3, 256), (6, 256, 6), (3, 512), (6, 384), (3, 768). Therefore, we take these five configurations to compare Transformer and PA-Transformer for different model capacities. From the figures, we can draw a conclusion similar to the machine translation dataset: the performance is not very sensitive to a small change of $\alpha$ and the proposed PA-Transformer model generalizes well for various model capacities.

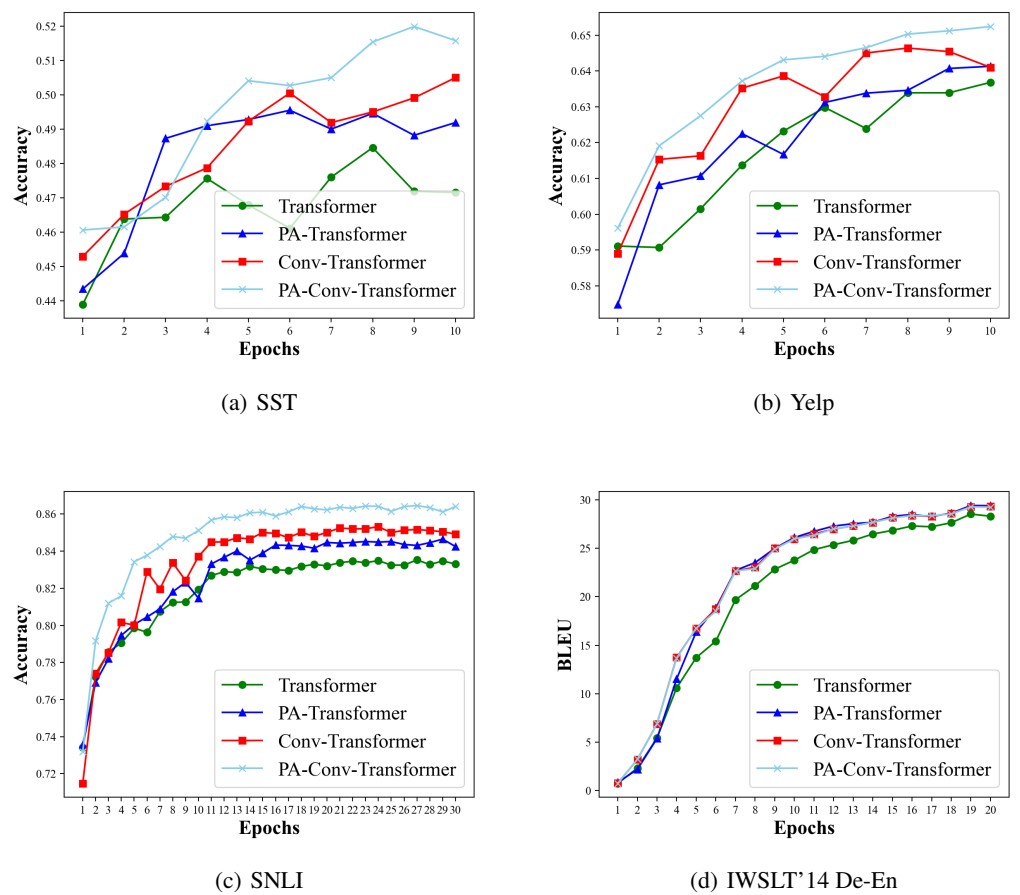

Figure 7: Learning curve comparison

## C.2 LEARNING CURVE COMPARISON

Figure 7 compares the learning curves of different models on multiple datasets, where the x-axis denotes epoch number and the y-axis denotes the test performance (BLUE score for IWSLT'14 De-En and accuracy for others). One can observe that PA-Conv-Transformer stabilizes the training process (especially for the smallest dataset, SST) and consistently outperforms other baselines at convergence.

## C.3 ATTENTION MAP VISUALIZATION

In order to get insight into the attention prediction mechanism, we visualize exemplar attention maps for both text and image inputs and find some interesting evidences.

### C.3.1 TEXT ATTENTION

We choose BERT-Base and PA-BERT-Base models for comparison on the CoLA dataset, a task for judging the grammatical correctness of a sentence. We select the sentence "*Mary tried John to go abroad.*" for visualization. Obviously, this sentence is grammatically wrong, and an effective model should capture the error part "*tried John to*" in order to give the true answer.

---

[9] https://github.com/tensorflow/benchmarks/tree/cnn_tf_v1.14_compatible
[10] https://github.com/leaderj1001/Attention-Augmented-Conv2d

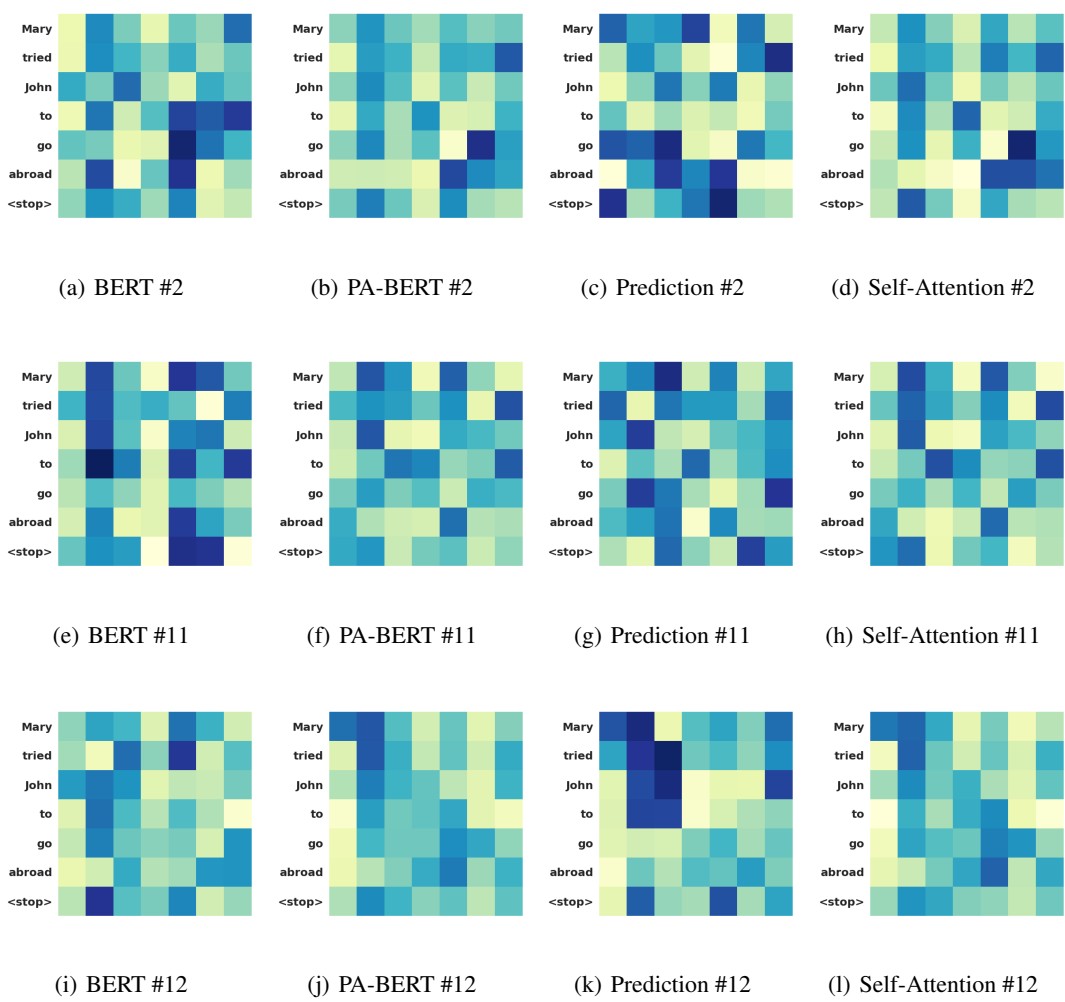

Figure 8: Attention maps of layer #2, #11 and #12 for "*Mary tried John to go abroad.*"

In Figure 8, we visualize related attention maps for three layers (#2, #11 and #12) in BERT-Base and PA-BERT-Base models. The second layer is the first layer that utilizes the input of the convolutional prediction model, and #11 and #12 are the last two layers. For each layer, we first show the attention maps from vanilla BERT and PA-BERT in the first and second columns respectively, then the predicted attention maps and supplementary self-attention maps are visualized in the third and fourth column. It should be noted that the second column is the linear fusion result of the third column and the fourth column according to Equation 4.

Consider layer #2, both BERT (Figure 8(a)) and PA-BERT (Figure 8(b)) pay major attentions on the verb phrase "*go abroad*". As shown in Figure 8(b), PA-BERT puts additional stress on the relation between word "*tried*" and the stop sign. This is reasonable because the stop sign is responsible of capturing sentence-level semantics and "*tried*" is a key word leading to the grammatical error. As we can observe in Figure 8(c), the attention on this part actually comes from the convolution-based prediction module, which is somewhat complementary to the self-attention map.

In order to ensure that the information obtained by the convolution is beneficial to the task, we visualize the last attention layer (#12) which is close to the classification output (see Figure 8(i-l)). In Figure 8(i), we can observe that BERT-Base still focuses on verbs and stop signs in the very last layer of transformer. The attention to the wrong phrase "*tried John to*" is still weak, which directly leads to a misclassification result for this case. In contrast, the attention scores between "*tried*" and

"*John*" become very high in PA-BERT (Figure 8(j)), largely owning to the predicted attention map illustrated in Figure 8(k).

We also visualize the attention maps of the #11 layer, which serves as the input of the #12 layer. To analysis the evolution of attention maps, we compare the difference between Figure 8(f) and Figure 8(k), as the latter is the output of convolutional prediction module by taking the former as input. We find that the convolutional prediction module helps to reason about the importance of word "*John*" based on the previous attention input. Specifically, it weaken the attention scores of the correct part and raises higher importance to the wrong part. As illustrated in Figure 8(k), the attention can be clearly seen in the upper left corner of the attention map where the error occurs. In this way, the error is fully captured in the final representation layer, assisting the model to generate a correct answer.

### C.3.2    IMAGE ATTENTION

In Figure 9, we compare the attention maps of Conv-Transformer and PA-Conv-Transformer for CIFAR-100 image classification. Compared to Conv-Transformer, our proposed method could capture better global information and at the same time emphasize the important local information. Specifically, the self-attention layer prefers to extract features from the global perspective, while the predicted attention highlights local features, which assists the self-attention mechanism to depict a more accurate outline. As shown by the visualized examples, Conv-Transformer fails to compute a explainable attention map. In contrast, with the help of attention prediction modules, PA-Conv-Transformer successfully identifies the objects in images.

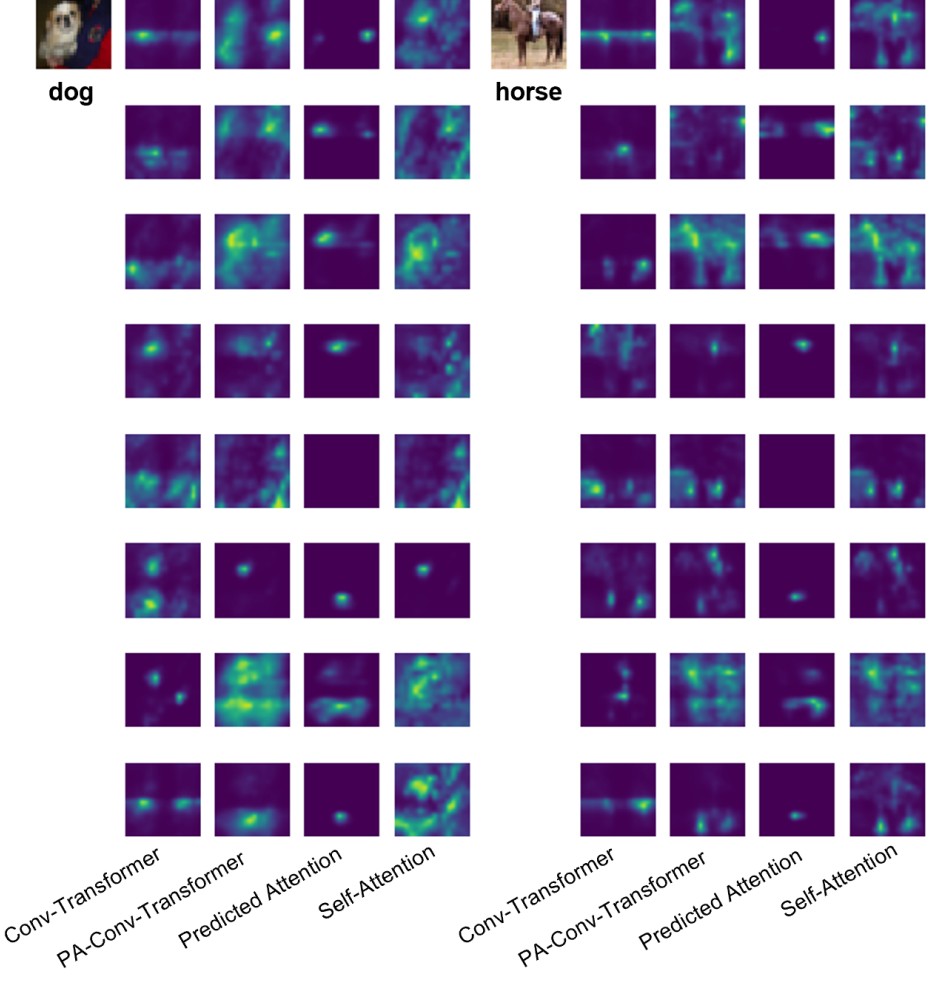

Figure 9: Attention maps of Conv-Transformer and PA-Conv-Transformer

