# OpenReview forum: "Predictive Attention Transformer: Improving Transformer with Attention Map Prediction"
_ICLR.cc/2021/Conference — Reject_

### Official Review · AnonReviewer2 · 2020-10-27
**Not convincing experiments and insufficient novelty.**

**Rating:** 2
**Confidence:** 4

**Review:**

The idea proposed in the paper is simple - "predict" future attention weights using past attention weights. A 2D CNN is used to mix previous N layer's attention maps. Strictly, this is also not "predicting" but instead generating". There is no supervised loss here.

The authors introduce a PA-Transformer model. The idea is to use previous attention maps to augment future attention weights. A stack of N previous attention weights is modeled with 2D CNN to generate future attention maps. The idea of predicting attention weights (or generating them) is not new (see https://arxiv.org/abs/2005.00743). The difference here is that there is a 2D CNN to model relationships between N previous layers. This is somewhat a pretty incremental extension of the Synthesizer-Transformer model.

There is also insufficient convincing evidence that using previous layer's attention to generate future attention weights is beneficial.

I think the experiments are lacking. The experiments on GLUE are only comparing against BERT (the least the authors could do is to compare side-by-side with at least a few other models). Machine translation datasets are tiny and ablation studies are unconvincingly run on SST and SNLI. The authors only run experiments using a preloaded checkpoint of BERT and do not apply their architecture to actually pretrain BERT which is also one weakness of this work. Hence, the paragraph beginning with "pretraining" is misleading". The results on GLUE are also weak and could be a result of variance over the existing BERT model.

The authors should also discuss how this can be implemented in a decoder setting since the current setup will disable causal attention.

Overall, I recommend a clear rejection. I think the key selling point and hypothesis behind this paper (using prev N layer attention) is not well supported. Experimental settings are also weak and there are insufficient convincing experiments to feel that this architecture is doing something useful.

---

> ### Author Response · Authors · 2020-11-17
> **Clarify the novelty and add more convincing results**
>
> Thanks for mentioning a related work, “Synthesizer”. It is a concurrent work to ours which is also in the reviewing process. More importantly, the key novelties of Synthesizer and our PA-Transformer model are different. The Synthesizer paper develops a strategy to synthesize attention maps and argues that explicit token-token interaction is not that important. However, in this paper, we argue that we can improve the effectiveness of inter-token relationships through convolutional neural networks with residual connections. This claim is well supported by the ablation study on multiple datasets. We also perform ablation experiments on large-scale ImageNet dataset to make the conclusion more convincing, and the results are updated in Table 6 in the latest version. We have empirically compared PA-Transformer with Synthesizer on the GLUE benchmark and verified the superiority of PA-Transformer.
>
> From the experimental perspective, we add more results to address your concerns (listed below). Hope these results can solve your concerns. We are open for more discussions if there are still questions or concerns from your side.
>
> 1. For the BERT-style fine-tuning experiments, we test more model backbones, including RoBERTa-Large and T5-base (see Table 2 in the revised version). The results show consistent improvement against vanilla baselines. In addition, the performance on T5-base is better than Synthesizer, although Synthesizer is pre-trained from scratch but PA-Transformer is only fine-tuned by the target task. This shows another practical advantage of PA-Transformer: getting superior performances without the need of pre-training from scratch again.
>
> 2. For the ImageNet dataset, we include additional results with more model capacities (ResNet-34, ResNet-101) in the revised version.  Our idea is proven to be general for both NLP and CV domains, so we can expect its impacts to more tasks and domains in the future. For Synthesizer, we are currently not sure if it is applicable to the computer vision domain.

---

> > ### Comment · AnonReviewer2 · 2020-11-22
> > **Questions**
> >
> > Thanks for conducting additional experiments.
> >
> > 1. Are the results on GLUE reported in the paper on the test set or the dev set?
> > 2. Are all GLUE tasks trained together or finetuned individually?
> > 3. I am confused about why Synthesizer not applicable to the vision domain?
> >
> > Thanks.

---

> > > ### Author Response · Authors · 2020-11-23
> > > **Answers**
> > >
> > > Thanks for your careful reading and helpful comments.
> > >
> > > 1 & 2.The results on GLUE reported in this paper are on the test set, and all tasks are fine-tuned individually. All models follow the same experimental protocol, except for Synthesizer. We check the descriptions in the Synthesizer paper and find that it co-trains all the tasks and reports the results on the dev set. Therefore, we also report the dev results on T5-Base architectures for a direct comparison. The average GLUE scores are listed below, with details updated in Table 2 in the latest version. PA-T5-Base outperforms Synthesizer-T5-Base on the dev set although it is only fine-tuned individually without benefiting from co-training of multiple tasks. We sincerely thank you for the comments which help us to find the discrepancy of experimental settings. Since we do not find the open source code of Synthesizer and the time is limited, we have to leave more controlled experiments to future work.
> > >
> > > T5-Base (Dev): 83.5
> > >
> > > Synthesizer-T5-Base (Dev): 84.1
> > >
> > > PA-T5-Base (Dev): 84.4
> > >
> > > -------------------------------
> > >
> > > T5-Base (Test): 83.2
> > >
> > > PA-T5-Base (Test): 84.5
> > >
> > > 3. Maybe we used a misleading word previously. Actually, Synthesizer can be applied to the computer vision domain. What we mean is that the original paper does not provide results in computer vision tasks, so we are not sure if a synthesized attention map still works well for tasks like image classification. It needs much time to implement a Synthesizer for image-related tasks and conduct sufficient experiments. Thus, a comparison in computer vision domain is considered as future work.
> > >
> > > Besides, the ablation results of residual connections and predictive convolutional layers on ImageNet are updated in Table 6 in the latest version.

---

> > > > ### Comment · AnonReviewer2 · 2020-11-23
> > > > **Thanks**
> > > >
> > > > Thanks,  i would like to see more controlled experiments  if the paper is accepted.
> > > >
> > > > I have raised my score to 5 for now.
> > > >
> > > > EDIT: I retracted my score increase (see below).

---

> > > > > ### Comment · AnonReviewer2 · 2020-11-24
> > > > > **Revising my score again**
> > > > >
> > > > > I read through the rebuttals, and my reviews again.
> > > > >
> > > > > The authors did not answer my concern about the attention being non-causal. And given that they run experiments of PA-T5-Base, it is also not clear what they did to the decoder.
> > > > >
> > > > > And given the limitation of this method being not able to autoregressive decode, it limits its impact. I am therefore moving my score back to a 3 unless the authors can ease my concerns about enabling causal attention.

---

> > > > > > ### Author Response · Authors · 2020-11-24
> > > > > > **Clarify the decoder**
> > > > > >
> > > > > > Thanks again for the detailed discussion! It is really helpful for us to improve this work.
> > > > > >
> > > > > > Here we add a clarification of the decoder part. In this paper, we focus on the improvement of encoders. All models use the same vanilla transformer for the decoder network (in T5 and machine translation experiments) to obtain a fair comparison. PA-Transformer has shown a solid improvement although only the encoder network is replaced. The idea of PA-Transformer is also applicable to the decoder network, which is considered as future work. Given consistent improvements on multiple tasks and domains for the encoder (including large-scale image classification), we think this work should be impactful in the current form. We will provide more experimental results with respect to decoder and Synthesizer in a later version.

---

> > > > > > > ### Comment · AnonReviewer2 · 2020-11-24
> > > > > > > **Not clear**
> > > > > > >
> > > > > > > My question was specifically directed to how the proposed method can be adapted for decoding. I can forsee issues in maintain causal attention and would like a clarification from the authors. Instead the authors replied to "our method can also be applied to decoding" without any substantiation nor explanation.
> > > > > > >
> > > > > > > Moreover, i think the fact that this paper doesn't use the proposed method on the decoder implies that there are complications in doing so. Otherwise this should have been evaluated holistically on both encoder and decoder for MT. For some reason, there also seems to be reluctance to discuss this issue.
> > > > > > >
> > > > > > > Given the number of transformer variants these days, a Transformer that cannot autoregressively decode will possibly find it difficult to make any impact. For this reason, I strongly recommend that this paper is rejected.

---

> > > > > > > > ### Author Response · Authors · 2020-11-25
> > > > > > > > **Answer your question about decoder**
> > > > > > > >
> > > > > > > > Dear reviewer,
> > > > > > > > Sorry, previously we might have a misunderstanding of your question and did not target it well. Here we explain how our method can be applied to a decoder.
> > > > > > > >
> > > > > > > > As the decoder cannot foresee succeeding tokens, we need a special mask. For example, when modeling the dependency  patterns for token i, we can not see the dependency pairs (i, i+1) or (i+1, i), but we can see the pairs (i, i-1), (i, i-2) or (i-1, i-2). These unmasked pairs serve as input to the convolution to help a better modeling of the dependency patterns around token i. For a vanilla transformer, we only need to mask (i, i+1). This is the major difference, and a way to solve the causal problem.
> > > > > > > >
> > > > > > > > We have implemented a version of this decoder, which got a marginal improvement (+0.1 BLUE) on De-En machine translation dataset compared to PA only encoder setting. We mainly select benchmarks for encoder evaluation, as the goal of this paper is to claim the contribution and improvement for text and image representation. We will provide extensive empirical study for the decoder in future works.
> > > > > > > >
> > > > > > > > At last, we really appreciate your time for the insightful discussions and suggestions no matter what your final decision is. Thank you very much!

---

### Official Review · AnonReviewer3 · 2020-10-28
**Adding convolution-based attention prediction module to Transformer to capture cross-layer dependencies**

**Rating:** 6
**Confidence:** 4

**Review:**


**Summary:**

This paper proposed a modification to the classical transformer architecture and demonstrated significant performance gain on multiple benchmark tasks in both natural language processing and computer vision. Specifically, the authors propose to introduce a convolution-based attention map prediction module, so the dependencies of attention maps across different layers can be captured. With the extensive experiments, the proposed modification is quite effective on improving the model's performance.


**Reasons for score:**

The idea of bridging attention maps across layers in Transformers is intuitive, as later layers can benefit from dependency structures learned from earlier layers. The model is validated on several benchmarks in both NLP and CV, and show some consistent improvements over baselines. However there are many unclear expressions and claims in the paper, and some ablations are missing which might be critical to better understand the module. Besides the paper definitely needs more proof-reading.


**Pros:**

1. The idea of treating multi-headed attention maps as multi-channel images is interesting, and the proposed convolution-based attention prediction module is a natural choice under such settings.
2. The proposed methods are validated with extensive experiments, and the performance gain is consistent and quite significant. This shows the effectiveness of the proposed approach.

**Cons:**

1. When applying transformer architecture to images, the image of shape $H \times W \times C$ is flattened as $X \in R^{N\times C}$ where $N=H \times W$. However for regular images the resolution is quite large, e.g., ImageNet is usually used in 224x224, then the N would be ~50k. In Section 4.2.1 it seems even for CIFAR it will be OOM (out of memory), but this is not mentioned for experiments with ImageNet in Section 4.2.2.
2. On Page 2 the authors claim that "... experimental results demonstrate the superiority of PA-Transformer in terms of accuracy, memory cost and computational efficiency ..." but I was only able to see the accuracy improvements; did I miss the experiments for computational efficiency? If it refers to the #Params and #FLOPs in e.g. Table 2, I'm actually curious: according to Table 1, Transformer and PA-Transformer has <0.01K (if not identical) FLOPs. Could you explain how this is calculated? Because if we count multiplications and additions as FLOPs, I think the PA module will definitely introduce more than 10 (=0.01K) FLOPs.
3. The authors claim several times (e.g. last sentence of Page 3) that self-attention module could "dedicate itself to incorporate layer-specific knowledge into *residual* attention maps". It seems arguable since in most cases the self-attention is dominating the generated attention map ($\alpha$ is usually small). Also, I'm wondering if the 0-layer PA in Table 5 corresponds to a direct skip-connection, i.e. simply copying the $A_{\text{pre-logits}}$ over to next layer. In fact I think it's a quite important ablation experiment, e.g. replacing "predicted attention" with "attention from previous layer", and may worth showing the results on other tasks too (e.g. on CIFAR and ImageNet).
4. Just curious: the $\alpha$ seems to have large variations across datasets, e.g. on CIFAR it is set to 0.01 but on SNLI and ImageNet it is set to 0.5. In addition to empirical validation results, is there any explanations for this?
5. In Table 1 and Table 2 since each experiments are replicated for five times, it's better to show the standard deviation (confidence interval) together with the mean value.

**Questions during rebuttal period:**

I've listed my questions in the cons section, and hopefully can be addressed during the rebuttal.


**Some typos and minor issues:**

-- The phrase "except for" is misspelled as "expect for" in several places (e.g., Page 5, 7th line of Section 4.1.1 "Models", Page 7, 2nd line of Section 4.2.1 Settings).
-- Multiple typos: "ResetNet-50" -> "ResNet-50", "Noe" -> "Note", etc. This paper needs more polishing.

---

> ### Author Response · Authors · 2020-11-17
> **Address the cons and make a revision accordingly**
>
> 1. For ImageNet experiments, we choose AA-ResNet (Attention Augmented Convolutional Networks, ICCV 2019) as the backbone model, which only applies attention augmentation on top layers to avoid OOM problems. PA-AA-ResNet strictly follows its architecture and only replaces the self-attention layers as PA-based ones.
>
> 2. Yes, the memory and computation costs are indicated by #Params and #FLOPS respectively. We calculate them by tensorflow code directly. (reference https://stackoverflow.com/questions/45085938/tensorflow-is-there-a-way-to-measure-flops-for-a-model). We appreciate you for pointing out the mismatch in #FLOPS! We also found this problem after paper submission and have fixed a bug in the code. The new metrics of flops have been updated in the revised version, and the conclusion is not affected.
>
> 3. Res-Transformer (0-layer PA) in Table 5 is exactly the ablation baseline for direct skip connection. The result proves the main claim in this paper: a residual connection is beneficial, and a chain of convolutional predictive layers brings additional advantages. We also perform ablation experiments on ImageNet and the results have been updated in the latest version.
>
> 4. Although the optimal alpha is small for some datasets, the performance lifts are obvious and stable. The value of alpha does not necessarily indicate the relative importance, but also covers some normalization issues. Originally we just utilized a standard residual connection. However, we find that the convolution-predicted attention sometimes has a sharp distribution, which harms the performance for some datasets. Therefore, we leverage an hyper-parameter “alpha” for the residual summation, which achieves a stable improvement for all datasets after hyper-parameter tuning on the validation set. In future work, we would like to investigate ways to get rid of this hyper-parameter while maintaining the superior performance.
>
> 5. We have updated Table 1 to include standard deviation of five runs, except for the machine translation experiments where we used a fixed seed following the baseline implementation. For Table 2, we run each experiment only once because the models are finetuned from pre-trained checkpoints and the performances are quite stable.
>
> Thanks again for pointing out some errors and typos in our paper. Your comments are really helpful for us to polish the paper content. We have corrected them in the revised version. Looking forward to having more discussions with you.

---

> > ### Comment · AnonReviewer3 · 2020-11-23
> > **Thanks for the clarification.**
> >
> > Thanks for the clarification, and for updating the main paper to address these concerns. I'm going to keep my rating for now (leaning towards acceptance), and will discuss with other reviewers later.

---

### Official Review · AnonReviewer1 · 2020-10-29
**unclear why the proposed method improves performance**

**Rating:** 6
**Confidence:** 4

**Review:**



This paper's main topic is the enhancement of Transformer models for improving performance in a task-independent way.


This paper first points out that the attention maps are trained independently among layers in the Transformer models.
The authors then hypothesize that the performance could be improved by integrating an additional module for estimating attention maps.
Therefore, they propose a method that yields the attention maps based on the lower layer's attention maps.

The experimental results on several different datasets from different domains show that the proposed method consistently improves the performance.
They are somewhat surprising results.


The following are the questions and concerns of this paper.


1,
The proposed method seems rather strange; why can the proposed method yield better attention map predictions?

I understand such a phenomenon if we provide the correct attention maps for model training.
However, it seems that the proposed method does not require any additional information on correct attention maps.
This is the largest mystery for me about this method.
Please elaborate on what architecture or mechanism enables the proposed model to provide better attention maps theoretically or empirically to support the authors' claim?
If I did not miss something, there are no clear explanations about it.



2,
the source of the effectiveness:
This additional module seems to also work as a sort of skip or short cut connections between layers.
In my feeling, the performance gain could be just the direct linking between attention mechanisms in each layer and not be caused by a better attention map prediction.
As a recent common knowledge in the community, the correct attention map can be, but not necessarily, a strong correlation to performance.
Please reveal the actual source of the performance gain to prove the correctness of the authors' claim.
Otherwise, there may be a risk of providing the wrong knowledge to the community.



3,
Related to the above two questions, this paper's main concern is that this paper does not provide more in-depth analyses of the proposed method that tell model behaviors or characteristics.
There are no intuitive and motivational examples of what kind of situation the proposed method successfully works.




4,
The current version does not have the "Conclusion" section, which most scientific papers have.
Of course, there is no rule that the paper always needs to have a Conclusion section.
However, I would like to know why the authors decided not to provide the Conclusion section.


I am willing to change my score if I got reasonable answers for all the questions and concerns written in the above reviews.

---

> ### Author Response · Authors · 2020-11-17
> **Explain motivations and provide in-depth analysis**
>
> 1. We think the improvement can be largely explained by two factors: (1) a residual connection to facilitate attention map learning; (2) a convolutional module to learn generalized patterns of inter-token relationship. In a vanilla Transformer, the self-attentions in each layer all learned independently.  Assembly, a residual connection on the attention map will help the succeeding layer to take the attention knowledge from previous layers. Moreover, there may be common patterns and transition rules shared across different attention maps. We hypothesis that a convolutional layer on a (n*n) attention map could leverage the locality of a relationship matrix and improves the performance of Transformer by generating better inter-term relationships explicitly. Using a convolutional induction bias is reasonable because nearby terms may have similar relationships. Table 5 shows empirical support of this explanation. Res-Transformer (0-layer PA) uses only residual connections, which already shows stable improvement on various datasets. Furthermore, adding one or two convolutional layers demonstrates additional benefits consistently.
>
> 2. As we have addressed above, residual connection is one of the reasons for performance improvement, but not all. As shown in Table 5, we empirically prove that the convolutional layers effectively learns a generalized function to produce better inter-token relationships based on existing attention maps.
>
> 3. We try to give more in-depth analysis through case studies for both text and image application.
> (1) In Figure 2, we visualize related attention maps for the last layer in BERT-Base and PA-BERT-Base models for a case of grammar check. The sentence is “Mary tried John to go abroad.” In Figure 2(a), BERT focuses on verbs and stop signs, leading to a misclassification. In contrast, PA-BERT learns to attend to the relationships between “tried” and “John”, which correctly captures the error part and gives a correct answer. The design of PA-Transformer to generalize inter-token relationships through convolutional inductive bias seems to be beneficial in this kind of scenario. More details are explained in Appendix C.3.1.
> (2) In Figure 1, we visualize a case of image classification. Again, PA-Conv-Transformer has a clearer attention map to show the horse than Conv-Transformer. As shown in the third column at Figure 1(b), the generated attention maps from convolutional modules highlights local key areas, so that self-attention could focus on complementary global information and collaboratively produce a better attention map. More details can be found in Appendix C.3.2.
>
> 4. Thanks for pointing out the miss of conclusion section. We have added more insight explanations, case study, and conclusion sections (some of them originally located in the appendix) in the 9-page main content. Hope this will make the paper more clear and solve your major questions and concerns. We are very pleased to have a further discussion with you if you have further comments or suggestions to our work.

---

> > ### Comment · AnonReviewer1 · 2020-11-23
> > **Official Blind Review #1**
> >
> > I really appreciate the authors' effort to answer my questions.
> >
> > I understand the fact that the proposed method improves the performance.
> > What I would like to see is clear theoretical or empirical evidence that supports the authors' claims.
> > I feel that the current rebuttal does not show such clear evidence but rather an intuition or guess.
> > There are many other choices for a similar extension, for example, using FFN instead of CNN or using a residual connection to different positions.
> > If such a similar extension gets similar improvements, then most of the explanations given by authors for the reasons for improvements might become wrong.
> > As I wrote in the review, I still feel that we need a more in-depth analysis to reveal the characteristics of the proposed method.
> >
> > I keep my score unchanged currently.
> > However, I also think this paper has a chance to be accepted to the conference.

---

> > > ### Author Response · Authors · 2020-11-25
> > > **More explanation and ablation results**
> > >
> > > Dear reviewer,
> > >
> > > Thanks for the helpful feedback! We fully understand your concerns, and would like to give a bit more explanation and ablation results.
> > >
> > > The most important argument in this paper is that an explicit modeling of token-token dependency patterns based on the previous attention map is useful. As we have shown in the ablation study, removing the convolutional module in attention prediction leads to a significant performance decay (see PA-Transformer v.s. Transformer with Residual Connection). Of course, one can use CNN, FNN or other models (even another self-attention layer) to capture the general dependency patterns in a 2D attention map. Because CNN is the most common choice for modeling 2D inputs, we leverage CNN in this paper for an initial attempt. We think the major contribution of this paper is to point out a new direction and encourage more advanced modeling of attention maps in the future.
> > >
> > > We agree with you that it would be more insightful to add more ablation settings. We show more ablation results on the SNLI dataset below (average of 5 runs). The empirical analysis on more datasets will be provided in the next version.
> > >
> > > Transformer: 83.22
> > >
> > > PA-Transformer: 84.63      (with 3*3 convolution kernel)
> > >
> > > Res-Transformer: 82.37
> > >
> > > This setting adds residual connections between adjacent transformer blocks.  It does not work, at least for this dataset.
> > >
> > > PA-Transformer only Residual Connection: 83.81
> > >
> > > This setting replaces CNNs in the PA-Transformer by direct residual connections. It performs better than vanilla Transformer, but decay obviously from PA-Transformer with 3*3 convolution kernel.
> > >
> > > PA-Transformer Synthesized Input: 83.21
> > >
> > > This setting uses a global trainable synthesized attention map as the input of PA-Transformer, instead of using the attention map of the previous layer.
> > >
> > > PA-Transformer FNN: 83.92
> > >
> > > This setting replaces CNNs in PA-Transformer by FNNs (analogous to 1*1 convolution). It has substantial improvement over vanilla Transformer, but decay relatively based on PA-Transformer with 3*3 convolution. This shows the advantage of capturing local dependency patterns with a 3*3 convolution kernel.
> > >
> > > PA-Transformer 5*5 CNN: 83.54
> > >
> > > This setting replaces 3 * 3 convolution with 5 * 5 convolution. The performance decay is perhaps due to the overfitting of a large kernel.
> > >
> > > As shown by the results above, using a dedicated model to capture attention patterns generally improves the performance, and 3*3 CNN achieves the best result on the SNLI dataset.
> > >
> > > Moreover, it should be noticed that the Conv-Transformer backbone model in our paper already adopts dense residual connections (similar to DenseNet) as described in the paper. Therefore, the improvement on top of Conv-Transformer truly comes from a better modeling of attention patterns. We will keep exploration along this direction and try to provide theoretical analysis in the future.

---

### Official Review · AnonReviewer4 · 2020-10-29
**Predictive Attention Transformer: Improving Transformer with Attention Map Prediction**

**Rating:** 6
**Confidence:** 4

**Review:**

This paper proposes a novel approach to improve self-attention through by bridging the attention maps from different layers via a chain of convolution-based prediction modules.  In particular, it proposes to augment the existing works on Transformer through supplementary prediction modules by CNN-based attention prediction layers.  The main contribution of this paper is the introduction of CNN-based attention prediction to enhance model predictions. Empirical studies are performed to show the superiority of the proposed model PA-Transformer over several SOTA approaches on NLP and image classification tasks.

Reasons for score:

 I like the idea of a chain of attention prediction to learn attention dependencies from the previous block. My major concern is about the clarity of the paper. Hopefully, the authors can address my concern in the rebuttal period.

Pros:

1. The paper addresses one of the most important issue of transformer: attention dependencies cross blocks or layers. For me, the problem itself is real and practical.
2. The proposed predictive attention transformer (PA-Transformer) is novel for capturing the attention dependencies transformer layers and address the problem of the self-attention maps learned independently for each layer. The design for using the PA-Transformer to tasks of NLP and image classification is reasonable and interesting.
3. This paper provides comprehensive experiments, including both NLP and image classification results, to show the effectiveness of the proposed framework.

Cons:

1. The paper claims that Multi-channel is one of the first works to take attention maps as multi-channel images for explicit modelling in the section of Introduction. It is better to clarity this point and give the difference between Multi-channel and multi-branch on method section.
2. Why does the paper call the proposed model as predictive attention? From my understanding, it is a type of residual connection for attention. Are the attention results used to predict some kind of tasks in intermediate layers?
3. Is the source codes available to reproduce the work?
4. It would be more convincing if the authors can provide a set of experiments about a baseline just using residual connection to bridge the layers, instead of CNN-based attention map prediction module, in the rebuttal period.

Questions during rebuttal period:

Please address and clarify the cons above

---

> ### Author Response · Authors · 2020-11-17
> **Clarify the cons and provide more results and analysis in the revised version**
>
> Thanks for your informative comments and suggestions for our work. We’d like to address your major concerns and questions as follows.
>
> 1. Multi-branch networks concatenate the output representations from transformer and convolutional encoders, where the convolution encoder takes text sequences as input. PA-Transformer, instead,  takes the n * n attention map as input for the convolutional layer, which is by design to model the general patterns of inter-term relationships explicitly. Through extensive experiments, we show that both designs are complementary and PA-Transformer still achieves significant improvement when applied to a multi-branch network architecture that already combines CNN and Transformer (see  PA-Conv-Transformer v.s. Conv-Transformer). The architecture of PA-Conv-Transformer is shown in Figure 4 in the appendix.
>
> 2. The goal of the convolutional layer is to learn a generalized function that generates better attention maps based on the previous knowledge. We do not use a predictive loss directly because there is no ground truth for the best attention map. Alternatively, we expect the convolutional layer to be guided by the task-specific loss indirectly. If there is enough data, this goal is achievable and the empirical results have proved this design.
>
> 3. We put the source code on Github using a new account for anonymity requirements. We have provided exemplar scripts for reproducing PA-BERT on GLUE benchmark and PA-AA-ResNet on ImageNet dataset.
> https://github.com/a-MLer/pa-transformer.
>
> 4. For baseline of residual connection, one can refer to Table 5 in the Analysis section, where Res-Transformer (equals to 0-layer PA) is the setting using only residual connections without convolutional layers. We can see the residual connection itself shows benefits as expected, while adding one or two convolutional layers further improves the performance by a large margin. This indicates that the convolutional module effectively learns the generalized pattern of inter-token relationships, which is the major contribution of this paper.

---

### Author Response · Authors · 2020-11-17
**Revision Summary**

1. We add evaluation results on RoBERTa-Large and T5-Base, which verify the superiority of PA-BERT on more pre-trained models. Impressively, PA-T5-Base outperforms Synthesizer on the same backbone model without the need of pre-training from scratch again. We also want to address that the conclusion of Synthesizer and our paper is not conflicting, but collaboratively confirming. Synthesizer argues that the current self-attention is not that useful, and a synthesized attention map even improve the performance for some cases.  PA-Transformer proposes a general strategy to improve the generalization ability of self-attention. We think there is a large chance that we can obtain a future improvement by composing Synthesizer and PA-Transformer together, and this sheds new light on a promising direction.

2. We add empirical results on ImageNet with multiple model capacities, including ResNet-34, -50 and -101. In addition, an ablation study for ResNet-34 on ImageNet is updated in the latest version (Table 6).

AA-ResNet-34           74.33

AA-ResNet-34 with PA-style Residual Connection        74.36

1-Layer PA-AA-ResNet-34                                                  74.90

2-Layer PA-AA-ResNet-34                                                  74.35

We can see that residual connections only improve the performance slightly, but adding one convolutional prediction layer brings much more benefit. To summarize, the experimental results on ImageNet prove the effectiveness of convolutional layers for predicting attention patterns.

3. The papers are polished following reviewers' suggestions. For short, we provide more insightful explanations, case studies, result numbers and formal conclusions.

---

### Decision · Program_Chairs · 2021-01-07
**Final Decision**

**Decision:**

Reject

**Comment:**

Multiple reviewers point out the interesting improvement to mix attention maps at different layers via convolution based prediction modules. This module is sufficient to show improvements only on encoder side while comparing to concurrent work Synthesizer.
However, the novelty of the work is limited as compared to other papers and the results though improved did not convince the reviewers fully to gain a strong accept.